# The Influence of Extraction Conditions on the Yield and Physico-Chemical Parameters of Pectin from Grape Pomace

**DOI:** 10.3390/polym14071378

**Published:** 2022-03-28

**Authors:** Mariana Spinei, Mircea Oroian

**Affiliations:** Department of Food Technologies, Food Production and Environment Safety, Faculty of Food Engineering, Stefan cel Mare University of Suceava, 720229 Suceava, Romania; m.oroian@fia.usv.ro

**Keywords:** grape pomace, pectin, conventional extraction, FT-IR, physico-chemical characteristics

## Abstract

Grape pomace is one of the most abundant by-products generated from the wine industry. This by-product is a complex substrate consisted of polysaccharides, proanthocyanidins, acid pectic substances, structural proteins, lignin, and polyphenols. In an effort to valorize this material, the present study focused on the influence of extraction conditions on the yield and physico-chemical parameters of pectin. The following conditions, such as grape pomace variety (Fetească Neagră and Rară Neagră), acid type (citric, sulfuric, and nitric), particle size intervals (<125 µm, ≥125–<200 µm and ≥200–<300 µm), temperature (70, 80 and 90 °C), pH (1, 2 and 3), and extraction time (1, 2, and 3 h) were established in order to optimize the extraction of pectin. The results showed that acid type, particle size intervals, temperature, time, and pH had a significant influence on the yield and physico-chemical parameters of pectin extracted from grape pomace. According to the obtained results, the highest yield, galacturonic acid content, degree of esterification, methoxyl content, molecular, and equivalent weight of pectin were acquired for the extraction with citric acid at pH 2, particle size interval of ≥125–<200 µm, and temperature of 90 °C for 3 h. FT-IR analysis confirmed the presence of functional groups in the fingerprint region of identification for polysaccharide in the extracted pectin.

## 1. Introduction

Pectin is a natural polymer found in the plant cell walls, mainly in fruits, such as apples, citrus fruits (oranges, lemons, etc.), and vegetables [1,2]. Pectin is a water-soluble dietary fiber consisting of esterified *D*-galacturonic acid (GalA) residues linked by α-(1,4) chain [3]. An essential aspect characterizing pectin is the degree of esterification of the uronide carboxyl group [4]. Commercial pectins are extracted from citrus peel, which contains 20–30% of pectin, apple pomace with 10–15% of pectin based on a dry matter, and alternative sources (sugar beet residue, sunflower heads, etc.) [5,6]. Generally, pectin is obtained by treating rough material with a hot aqueous acid solution at pH about 2 [7]. The extraction time of pectin depends on the raw material, type of solvent, solute to solvent ratio, pH of solution, temperature, calcium ion concentration, particle size, etc. [8,9].

Presently, other vegetables and fruits residues represent a new source of pectin with impressive food utilization as valorizing innovative products and as functional component with health-promoting benefits [10]. Currently, many non-conventional sources of pectin, such as cocoa husks, watermelon rinds, pumpkin waste, tomato waste, mango peels, banana peels, etc. were studied [11,12,13,14,15]. The wine industry is one of the greatest agricultural activities which generates different by-products (e.g., grape pomace, which is consist of skins, seeds, and bunches; lees sediments; vinasse etc.) [16]. A lot of studies have reported that winery residues could be a useful substitute with great potential to develop many bioproducts [16,17]. Moreover, the valorization of winery by-products (gaseous, liquid, or solid) promotes to identify a sustainable solution with value-added products in biorefinery management, maintaining the circular economy and environment [16,17,18,19].

Based on these remarks, we proposed in present study the application of grape pomace of two different *Vitis vinifera* varieties (Fetească Neagră and Rară Neagră) as an unconventional source for pectin extraction. Fetească Neagră is a dark-skinned grape variety cultivated mainly in several Romanian regions and also in the Republic of Moldova. The Fetească Neagră grapes are medium to large, cylindrical-conical with spherical, medium-sized berries, dark purple skins and have a weight of 190–230 g [20]. The Rară Neagră grapes are flattened, or spherical, medium-sized berries, ruby colored skins and weigh 200–240 g [20].

Concerning the extraction of pectin from grape pomace, only a few studies were found in the literature [21,22,23]. Due to the lack information on extraction of pectin from grape pomace of Fetească Neagră and Rară Neagră varieties and the need to find new sources of pectin, we aimed to study the influence of the acid type (organic vs. mineral), temperature, pH, particle size, and time on the yield and physico-chemical parameters (galacturonic acid content, degree of esterification, methoxyl content, molecular, and equivalent weight) of pectin from grape pomace of two different *Vitis vinifera* varieties (Fetească Neagră and Rară Neagră) using conventional method of extraction and establish the optimal conditions of grape pomace pectin extraction. Moreover, the other objective of this study was to define the structure of pectin samples using FT-IR analysis and scanning electron microscopy (SEM).

## 2. Materials and Methods

### 2.1. Materials

Grape pomace was collected by processing two different *Vitis vinifera* varieties (Fetească Neagră and Rară Neagră) from a 2019 harvest, cultivated in the Bugeac area, Republic of Moldova. The grape pomace was dried in an oven with air circulation Zhicheng ZRD-A5055 (Zhicheng, Shanghai, China) at 50 °C until constant weight was achieved. Then, dried pomace was powdered using a food processor (KitchenAid, Benton Harbor, MN, USA) and separated it on the following particle size intervals: <125 µm, ≥125–<200 µm and ≥200–<300 µm using an analytical sieve shaker Retsch AS 200 Basic (Retsch GmbH, Haan, Germany).

Nitric acid, sulfuric acid, citric acid, sulfamic acid, hydrochloric acid, sodium hydroxide, *m*-hydroxydiphenyl, potassium hydroxide, sodium tetraborate, *D*-galacturonic acid, and ethyl alcohol were acquired from Sigma-Aldrich (Darmstadt, Germany).

### 2.2. Methods

#### 2.2.1. Extraction and Purification of Pectin from Dried Grape Pomace

Initially, a sample of 10 g grape pomace powder was mixed with 100 mL of solvent (solid-liquid ratio of 1:10, *w*/*v*) acquired by adding nitric acid, sulfuric acid, and citric acid to ultrapure (Milli-Q) water until a pH 2 was achieved. For each acid, three different mixtures were prepared according to the particle size intervals. Then, the mixtures were kept in a weather bath Precisdig (JP Selecta, Barcelona, Spain) at the temperature of 90 °C for 3 h.

After extraction, the mixtures were cooled to room temperature, around 20–22 °C. Firstly, the solid material was segregated by centrifugation at 2320 g and 22 °C for 35 min. Then, the obtained supernatants were put through a clean strainer and placed into the neck of a Duran^®^ laboratory glass bottle with pouring ring and screw cap. Afterwards, ethyl alcohol (>96%, *v*/*v*) was added to supernatants in order to achieve 1:1 ratio (*v*/*v*). The mixtures were kept at 4–6 °C for 12 h to accomplish the precipitation. The precipitated pectin was separated by centrifugation at 2320 g and 22 °C for 30 min. The pectin was washed 3 times with ethyl alcohol (>96%, *v*/*v*) and dried in an oven with air circulation Zhicheng ZRD-A5055 (Zhicheng, Shanghai, China) at 50 °C until constant weight was achieved.

#### 2.2.2. Pectin Yield

Pectin yield was calculated using Equation (1):(1)$$\mathrm{Yield}\;\left(\%\right)=\frac{m_{0}}{m}\times 100\;$$
where: $m_{0}$—weight of dried pectin (g), m—weight of dried grape pomace powder (g) [5,24].

#### 2.2.3. Galacturonic Acid Content

The galacturonic acid content (GalA) of pectin was estimated using the sulfamate/*m*-hydroxydiphenyl method developed by Filisetti-Cozzi and Carpita [25] and Melton and Smith [26]. Sample preparation was made according to Miceli-Garcia [27]. 20 mg of dry pectin were added to 50 mL of Milli-Q water at 40 °C and mixed using a magnetic stirrer (Heidolph Instruments GmbH & Co. KG, Schwabach, Germany) until samples were completely dispersed. Then, the volumes were adjusted to 100 mL with Milli-Q water at 40 °C. Aliquots of 400 µL from the pectin solutions were placed in glass tubes, followed by addition of 40 µL of 4 M potassium sulfamate solution at pH 1.6 and vigorously mixed using a laboratory shaker (Heathrow Scientific, Vernon Hills, IL, USA) for at least 5 s. Afterwards, 2.4 mL of sulfuric acid containing 75 mM of sodium tetraborate was added and mixed using a laboratory shaker. The mixtures were placed in a boiling water bath for 20 min and then cooled in an ice bath for 10 min. Right away, after cooling, 80 µL of *m*-hydroxydiphenyl solution in 0.5% sodium hydroxide (*w*/*v*) was added and vortex mixed for 5 s. The absorbance was read at 525 nm against the reagent control using a UV-Vis-NIR spectrophotometer (Shimadzu Corporation, Kyoto, Japan).

A *D*-galacturonic acid calibration curve was prepared for each batch of samples.

#### 2.2.4. Degree of Esterification

The degree of esterification (DE) of pectin was determined by the titrimetric method described by Franchi [28] and Wai et al. [29], as follows: 50 mg of pectin were moistened with 10 mL of boiled Milli-Q water. After complete dissolution of pectin, the sample was titrated with 0.1 mol/L sodium hydroxide ($V_{1}$) using phenolphthalein as indicator. Then, 20 mL of 0.5 M sodium hydroxide were added, and the solution was kept under continuous stirring at 400 rpm for 30 min in order to achieve saponification. Afterwards, 20 mL of 0.5 M hydrochloric acid were added to neutralize the solution and titrated with 0.1 N sodium hydroxide ($V_{2}$). The degree of esterification was calculated with the Equation (2):(2)$$\mathrm{DE}\;\left(\%\right)=\frac{V_{2}}{V_{1}+V_{2}}\times 100$$
where: $V_{1}$—volume of sodium hydroxide used for the first titration (mL), $V_{2}$—volume of sodium hydroxide used for the second titration (mL).

The DE of pectin samples was measured in triplicate.

#### 2.2.5. Equivalent Weight

The equivalent weight (EW) of pectin samples was determined by Ranggana [30], as follows: 0.25 g of pectin were completely dissolved in 100 mL Milli-Q water under continuous stirring at 300 rpm for 1 h. Then, 1 g of sodium chloride and 5 drops of phenolphtalein as indicator were added. The solution was titrated with 0.1 N sodium hydroxide until the color changed to pink and persisted for at least 30 s. The EW was calculated using Equation (3):(3)$$\mathrm{EW}=\frac{1000\times m}{V\times N}$$
where: m—weight of sample (g), V—volume of alkali (mL), N—normality of alkali.

The EW of pectin samples was measured in triplicate.

#### 2.2.6. Methoxyl Content

The neutral solution was collected from the determination of EW and 12.5 mL of 0.25 M sodium hydroxide were added in a stoppered flask. The mixed solution was stirred thoroughly and kept for 30 min at room temperature around 20–22 °C. Then, 12.5 mL of 0.25 M hydrochloric acid was added and titrated with 0.1 N sodium hydroxide to the same end as before [30]. The methoxyl content (MeO) was calculated using Equation (4):(4)$$\mathrm{MeO}\;\left(\%\right)=\frac{V\times N\times 3.1}{m}$$
where: V—volume of alkali (mL), N—normality of alkali, m—weight of sample (g).

The MeO of pectin samples was measured in triplicate.

#### 2.2.7. Molecular Weight

Molecular weight (Mw) was determined by high-performance size-exclusion chromatography using a HPLC system (Shimadzu Corporation, Kyoto, Japan) equipped with a LC-20 AD liquid chromatograph, SIL-20A auto sampler, a Yarra 3 µm SEC-2000 column (300 × 7.8 mm; Phenomenex, Torrance, CA, USA) and coupled with a RID-10A refractive index detector (Shimadzu, Kyoto, Japan). The samples were made according to Dranca et al. [31]. Elution was carried out with 0.1 M sodium nitrate solution containing 0.024% sodium azide at a flow rate of 0.5 mL/min and temperature of 25 °C. Pectin solutions (0.3% *w*/*w*) were filtered through 0.45 μm membranes before injection. The calibration was performed using pullulan standards (Shoko Science Co., Tokyo, Japan). The LC solution software version 1.21 (Shimadzu Corporation, Kyoto, Japan) was utilized to collect the data.

#### 2.2.8. Color

The color of the pectin samples was analyzed in triplicate at 25 °C with a CR-400 chromameter (Konica Minolta, Tokyo, Japan). CIE L*; hue (h*_ab_) and chroma (C*_ab_) were obtained from the reflection spectra of the samples with illuminant D65 and 2° observer.

#### 2.2.9. FT-IR Analysis

Fourier-transform infrared spectroscopy (FT-IR) analysis was made using a Nicolet i-20 spectrophotometer (Thermo Scientific, Karlsruhe, Dieselstraße, Germany). The spectra were recorded in transmission mode using the attenuated total reflectance (ATR) system within the wave number range of 4000–400 cm^−1^ at a resolution of 4 cm^−1^. The Omnic software (Version 9.9.473, Thermo Fisher Scientific, Waltham, MA, USA) was used to display the spectra. The samples were placed on the ATR crystal, and the spectra were recorded in triplicate.

#### 2.2.10. Microstructure

The microstructure of the pectin samples was examined by scanning electron microscopy (SEM; SU-70, Hitachi, Tokyo, Japan). Dried pectin powder was fixed to the sample table with conductive double-sided adhesive carbon tape and analyzed using an accelerating voltage of 30 kV at different magnifications (300×, 500× and 1000×). The Vega software (Version 3.5.2.1, Tescan Orsay Holding, Brno, Czech Republic) was utilized to display the structural morphology.

#### 2.2.11. Statistical Analysis

Results were submitted to analysis of variance (ANOVA) using XLSTAT software (Addinsoft, New York, NY, USA). The ANOVA test was used to evaluate the difference between means at the 95% confidence level (*p* < 0.05) with Fisher’s least significant difference (LSD) procedure.

## 3. Results and Discussion

First, an important remark must be made. In order to optimize the extraction procedure, the following conditions were established: acid type, particle size intervals, pH, time, and temperature (Figure 1). The maximum pectin yield was obtained after 3 h extraction with citric acid for ≥125–<200 µm of particle size (Table 1).

Thus, the influence of pH, time, and temperature on the yield and physico-chemical parameters of pectin from Fetească Neagră and Rară Neagră grape pomace were analyzed for the extraction with citric acid and ≥125–<200 µm of particle size.

### 3.1. Influence of Grape Pomace Variety on the Yield and Physico-Chemical Parameters of Pectin

The yield and physico-chemical characteristics of pectin depend on the source, extraction method, and different factors used (solid-to-liquid ratio, particle size, pH, temperature, and extraction time) [32,33]. The influence of grape pomace variety on the yield and physico-chemical parameters of pectin from grape pomace using acid extraction, pH 2, and temperature of 90 °C for 3 h is presented in Table 1. The yield of pectins extracted from the Rară Neagră (RN) and Fetească Neagră (FN) grape pomaces varied between 5.70% and 6.07%, respectively. Colodel et al. [23] obtained the maximum pectin yield of 11.1% under the optimized conditions (pH 2.08 for 135.23 min with a solid-to-liquid ratio of 35.11 mL/g) from Chardonnay grape pomace. The EW presented a value of 548.09 g/mol for pectin from RN grape pomace as compared to the value of FN grape pomace pectin (555.38 g/mol). Similar results of EW were obtained by [15] for citrus peel pectin (577 g/mol) and apple pomace pectin (551 g/mol) using citric acid extraction (solid-to-liquid ratio 1:20). Minjares-Fuentes et al. [21] achieved a value for EW of 163.9 kDa for Cabernet Sauvignon grape pomace pectin using an ultrasound-assisted treatment with citric acid (pH 2, 75 °C for 60 min). The obtained values of DE and M_w_ were not significantly influenced by grape pomace variety (*p* > 0.05).

Additionally, Limareva et al. [22] did not find a high difference of DE values among seven grape pomace varieties, DE ranging from 52 to 65%.

The physico-chemical parameters, such as GalA, MeO, and color measuring parameters (L* and h*_ab_) were affected by the grape pomace variety (*p* < 0.0001). Galacturonic acid and methoxyl content define the quality of pectin [32,34]; their values in pectin can affect the structure, composition, and texture of the pectin gel formed [35,36]. The GalA presented the following values, 33.14 g/100 g and 48.47 g/100 g for FN and RN grape pomace pectin, respectively, but Colodel et al. [23] obtained an amount of 56.8% of GalA for Chardonnay grape pomace pectin by acid extraction. The MeO was 4.30% and 5.55% for RN and FN grape pomace pectin, respectively. Since MeO was below 7% for all samples, the extracted pectin from FN and RN grape pomaces was a low ester characteristic and was considered as being “desirable” in terms of quality [31]. Therefore, pectin with low MeO will form a thermo-irreversible gel [31]. Pectin from RN grape pomace had the highest lightness (L*) value (49.14), as well as higher hue (h*_ab_) value (39.27) and chroma (C*_ab_) value (11.26) in comparison with FN grape pomace pectin (41.62, 31.06, and 10.89 for L*, h*_ab_, and C*_ab_, respectively). Thus, pectin samples were presented a color ranging from red to red-purple according to CIE chromaticity diagram. Grape pomace extracted pectin color was predominantly due to tannins and anthocyanins, which are the main polyphenolic compounds responsible for color in red grape cultivars [37]. During the conventional extraction process, unbinding of cell wall makes the dissolution of pigments in acidified water. These color components become captured in pectin during the precipitation phase. This might be the reason for the color parameters values. In order to approve the results obtained for color parameters (L*, h*_ab_ and C*_ab_), some relevant pictures of pectin samples extracted under the influence of acid type with fixed conditions: pH 2, particle size of ≥125–<200 µm, 90 °C, and extraction time of 3 h are showed in Figure 2.

### 3.2. Influence of Acid Type on the Yield and Physico-Chemical Parameters of Pectin

The use of mineral acids (sulfuric, nitric, hydrochloric, etc.) for pectin extraction has been reported to environmental consequences and increased costs [33,38]. Regarding the developing concept of “green chemistry and technology” and disadvantages related to using of mineral acids, the focal point is the utilisation of organic acids (acetic, citric, etc.) for pectin extraction [33]. The influence of acid type on the yield and physico-chemical parameters of pectin from grape pomace using acid extraction, pH 2, temperature of 90 °C for 3 h is presented in Table 1. The pectin yield, DE, MeO, and color parameters (h*_ab_ and C*_ab_) were not significantly influenced by acid type (*p* > 0.05). Thus, the pectin yield presented the following values, 5.80%, 5.84%, and 6.01% for sulfuric, nitric, and citric acid extraction, respectively. However, Raji et al. [39] reported a high difference for yield of melon peel pectin among acid type (sulfuric, phosphoric, nitric, lactic, acetic, hydrochloric, citric, and tartaric) of extraction with a range from 1.4 to 25.3%. MeO had an amount of 4.56% for nitric acid, 4.89% for sulfuric acid and 5.31% for citric acid. Additionally, Sengkhamparn et al. [40] did not find a high difference of MeO values from tomato waste pectin, 21.45%, 22.61%, and 22.67% for nitric, hydrochloric, and citric acid extraction, respectively. The DE ranged from 73.19 to 74.15%. However, Yapo [38] established that acid type affects the DE of pectin from yellow passion fruit rind. They obtained the highest value of DE (73%) for 0.01 M citric acid extraction and the lowest value of DE (29%) for 0.03 M nitric acid extraction.

The GalA and EW were significantly influenced by the acid type (*p* < 0.01). Therefore, the GalA presented the following values: 31.45 g/100 g for sulfuric, 38.95 g/100 g for nitric, and 52.01 g/100 g for citric acid of pectin from grape pomace. As well, Ma et al. [41] reported that GalA is determined by acid type of pectin extraction. They obtained a range 52.7–72.4 g/100 g of GalA for lactic acid, 60.8–77.8 g/100 g for malic acid and 60.7–78.8 g/100 g for citric acid extraction of pectin from beet pulp. The EW varied from the lowest value (531.04 g/mol) for nitric acid to the highest value (566.85 g/mol) for citric acid extraction of pectin from grape pomace. Thus, Khan and Nandkishor [42] confirmed that acid type influence the EW of pectin from wild plums (*Prunus domestica*). Considering the whole range, citric acid and hydrochloric acid extraction exhibited the highest EW (1020 g/mol) and the lowest EW (833.3 g/mol), respectively [42]. Additionally, M_w_ was significantly influenced by the acid type (*p* < 0.0001). The M_w_ ranged from 5.25 × 10^4^ g/mol for nitric acid to 5.34 × 10^4^ g/mol for citric acid. The highest value of L* (48.33) was obtained by pectin extraction with citric acid, while the lowest L* (40.44) was obtained by nitric acid extraction. This can be explained by the main phenolic compounds, such as phenolic acids which are found in grape pomace and retained after extraction. Thus, Manasa [43] recovered free (epicatechin, rutin, and ferulic acid) and bound (rutin, epicatechin, and gallic acid) polyphenols from coffee pectin after acid extraction.

Therefore, pectin extraction with organic acid can be applied to generate functionality-enhanced pectin. The organic acid extraction of pectin is opportune; apart from high extraction yield, the obtained pectin is increased in galacturonic acid [42]. It known that mineral acids (nitric and sulfuric) are highly corrosive, which may be a threat to human health (e.g., skin and eyes), in contrast with citric acid, which is better for environment and “green” economy [40].

### 3.3. Influence of Particle Size on the Yield and Physico-Chemical Parameters of Pectin

The particle size increases the extraction efficiency and improves the functional properties of pectin, such as solubility, gelation, emulsification, texture, and viscosity [44,45]. The influence of particle size on the yield and physico-chemical parameters of pectin from grape pomace using acid extraction, pH 2, and temperature of 90 °C for 3 h is presented in Table 1. Thus, the yield, GalA, MeO, and M_w_ of pectin were significantly influenced by the particle size of grape pomace (*p* < 0.0001). The grape pomace pectin yield presented the following values, 4.64%, 5.71%, and 7.31% for particle size intervals of <125 µm, ≥200–<300 µm and ≥125–<200 µm, respectively. Huang et al. [44] reported that pectin yield from sugar beet pulp was ranged from 15.81% (406 µm) to 20.50% (25 µm). This can be explained by breaking the cell walls of plant matrix [46]. Additionally, Huang et al. [44] presented an increase of GalA from 38.51 (406 µm) to 59.97% (25 µm). Similar results were obtained for pectin extraction from mango peel, the yield was significantly higher for the particle size of 42 μm [45]. Geerkens et al. [45] reported that these results were influenced by the mango peel processing (blanching, drying, and irradiation); this may be an explanation for the different influence of the particle size on physico-chemical parameters of pectin. Therefore, the highest values of GalA (56.33 g/100 g) and M_w_ (5.35 × 10^4^ g/mol) from grape pomace pectin was obtained for particle size interval of ≥125–<200 µm. The MeO of pectin from grape pomace showed the following values, 3.98%, 4.60%, and 6.18% for particle size intervals of ≥200–<300 µm, <125 µm, and ≥125–<200 µm, respectively.

The DE, EW, and color parameters (L*, h*_ab_, and C*_ab_) were not significantly influenced by particle size (*p* > 0.05). The DE and EW of pectin from grape pomace presented a range of 71.31–74.92% and 540.38–565.70 g/mol, respectively. Geerkens et al. [45] reported the highest content of the degree of methylation (66.4%) from mango peel pectin for a 0.25 mm sieve in comparison with a 0.5 mm sieve (53.3%). The highest value of L* (48.33) and h*_ab_ (36.80) was obtained by pectin extraction with <125 µm particle size, while the lowest L* (40.44) and h*_ab_ (34.08) by ≥200–<300 µm of grape pomace. Thus, pectin samples from grape pomace were presented a color ranging from red-orange to red according to CIE chromaticity diagram.

### 3.4. Influence of pH on the Yield and Physico-Chemical Parameters of Pectin

The strength of mineral or organic acid used to extract pectin from different fruits and vegetables had significant effect on the pectin yield. Moreover, the yield of pectin decreased with the increasing of acid strength [29,46,47]. The influence of pH on the yield and physico-chemical parameters of pectin from grape pomace using citric acid extraction, ≥125–<200 µm particle size, and temperature of 90 °C for 3 h is presented in Table 2. Thus, the yield, DE, EW, MeO, M_w_, and color parameters (L* and C*_ab_) were significantly influenced by the pH of acid solution (*p* < 0.0001). The grape pomace pectin yield presented the following values, 6.14%, 7.56%, and 12.43% for pH of 3, 2, and 1, respectively. The similar results were obtained by Yapo [38] in order to present the effect of acid extractant nature on the biochemical characteristics of pectin from yellow passion fruit rind. They determined that nitric acid at pH value of 1.4 was the optimal acid solvent for 12.8% pectin yield.

However, Colodel and Petkowicz [48] reported that the highest pectin yield (15.6%) from cubiu fruit peel was obtained at pH 2 and the lowest (4.5%) at pH 1 under the influence of the extraction conditions, boiled in nitric acid for 4 h. The DE and M_w_ of pectin from grape pomace presented a range of 70.39–78.92% and 5.28 × 10^4^–5.39 × 10^4^ g/mol, respectively, for different pH values. The higher the pH, the higher the value of DE of pectin. Additionally, Yapo [38] established that DE increased significantly from 56% to 70% as pH increased from 1.8 to 2.5. Opposed, Kalapathy and Proctor [47] reported that the DE of soy pectin measured by FT-IR, was from 53% to 60%, but with no statistically significant differences, and pH 2 of 2-propanol did not have a significant influence on DE. The value of EW of pectin from grape pomace ranging from 527.51 g/mol for pH 3 to 582.97 g/mol for pH 2. The enhancement or decrease of EW of pectin depends upon the quantity of free acid [34]. Wathoni et al. [32] related a high value of EW (6330.76 g/mol) and MeO (2.86%) for mangosteen rind pectin, which was obtained under the following conditions of extraction: acidified water with sulfuric acid at pH 2 in 90 °C for 120 min. The MeO of grape pomace pectin presented a range of 6.53–6.99%. The GalA was significantly influenced by the pH of acid solution (*p* < 0.01). The GalA of pectin presented the following values: 39.40 g/100 g, 61.67 g/100 g, and 45.57 g/100 g for pH of 1, 2, and 3, respectively. Yapo [38] reported a GalA ranging from 68% to 72% at pH 3.5. Hence, at pH 3.5, pectin is in a 50% ionized state, because of different balance between hydrophilic and hydrophobic character [49,50]. The color parameters (L*, h*_ab_ and C*_ab_) were influenced by pH value of extraction pectin from grape pomace. The highest values of L* (59.92), h*_ab_ (42.48), and C*_ab_ (12.37) were obtained at pH 3 and the lowest values of L* (23.89), h*_ab_ (25.06), and C*_ab_ (4.41) at pH 1. This can be explained by the fact that the stability of polyphenols is pH-dependent; the lower the pH, the more stable the polyphenols during extraction.

### 3.5. Influence of Time on the Yield and Physico-Chemical Parameters of Pectin

The increasing time extraction using citric acid is correlated with an enhancement of the pectin yield [51]. The influence of time on the yield and physico-chemical parameters of pectin from grape pomace using citric acid extraction, ≥125–<200 µm particle size, pH 2, temperature of 90 °C is presented in Table 3.

Thereby, the yield, DE, M_w_, L*, and h*_ab_ were significantly influenced by the time of extraction (*p* < 0.0001). The pectin yield presented the following values, 5.38%, 5.77%, and 7.50% for 1, 2 and 3 h, respectively. Thus, the results indicated that the yield of grape pomace pectin increased with enhancement of extraction time. Kulkarni and Vijayanand [52] studied the influence of extraction conditions on the quality characteristics of pectin from passion fruit peel; they obtained the highest yield of pectin of passion fruit peel from sum of two extractions (14.83 g/100 g) for 60 min at pH 2, 98.7 °C, and solid-to-liquid ratio 1:30 (*w*/*v*). Additionally, they reported that pectin extracted for 30 min had a higher MeO (9.84 g/100 g) and EW (839.1); in this case, MeO and EW decreased with increase of extraction time. Irrespective of these results, the highest value of MeO (6.54%) and EW (618.42 g/mol) were obtained using citric acid at pH 2, for 3 h of extraction for the pectin from grape pomace. The highest DE of pectin from grape pomace (77.77%) was obtained at extraction time of 3 h and the lowest for 1 h (67.17%). Chan and Choo [51] reported the highest degree of methylation of pectin from cocoa husks (57.86%) using citric acid at pH 2.5 at 50 °C for 1.5 h. The M_w_ ranging from to 5.25 × 10^4^ to 5.38 × 10^4^ g/mol at pH 2.

The GalA was significantly influenced by the time of extraction (*p* < 0.001). Thus, the GalA presented the following values: 33.69 g/100 g, 39.04 g/100 g, and 46.98 g/100 g for 1, 2, and 3 h, respectively. Kulkarni and Vijayanand [52] obtained a content of anhydrogalacturonic acid (88.2 g/100 g) of pectin from passion fruit peel. The time of extraction influenced the color parameters (L* and h*_ab_), the highest values of L* (71.52), and h*_ab_ (46.71) were presented for grape pomace pectin extracted with citric acid pH 2, at 90 °C for 3 h. Berardini [53] reported an increase of L* and hue angle of the purified extracted of the pectin from mango peel. In contrast, they established that there were no differences for the pectin obtained before and after adsorption of the polyphenols.

### 3.6. Influence of Temperature on the Yield and Physico-Chemical Parameters of Pectin

Extraction temperature is a major characteristic affecting the structure of the pectin. The increasing temperature significantly enhanced the yield of pectin using citric acid extraction [34,51,54]. The influence of temperature on the yield and physico-chemical parameters of pectin from grape pomace using acid extraction, ≥125–<200 µm particle size, pH 2, for 3 h is presented in Table 4. Thus, the yield, GalA, EW, MeO, and C*_ab_ were significantly influenced by the temperature of extraction (*p* < 0.0001). The grape pomace pectin yield presented the following values: 4.23, 6.02, and 7.30% for 70, 80, and 90 °C, respectively. Chan and Choo [51] extracted pectin from cocoa husks using three types of extractants (water, citric acid at pH 4 or 2.5, hydrochloric acid at pH 4 and 2.5) with a specific solid to liquid ratio (1:10 and 1:25) incubated at 50 or 90 °C and time (1.5 or 3 h). They obtained the highest yield of pectin from cocoa husks (5.66%) at 95 °C using citric acid at pH 2.5 and the lowest (3.38%) at 50 °C using water as extractant [51]. Chen et al. [1] studied the influence of temperature as a decisive factor for the properties of *Citrus unshiu* fruits pectin. They reported that temperature below 40 °C kept the pectin structure more intact. The highest value of GalA was obtained at 90 °C (62.21%). Contrastingly, Chan and Choo [51] established the highest yield of uronic acids from cocoa husks pectin (59.36%) at 50 °C using citric acid at pH 2.5.

A lower temperature caused less degradation to pectin structure. Moreover, the highest values of EW, MeO, and M_w_ of pectin extracted from grape pomace were 608.30 g/mol, 6.75%, and 5.32 × 10^4^ g/mol, respectively, for time extraction of 90 °C. The DE was not significantly influenced by the temperature and presented a range of 74.65–81.40% (*p* > 0.05). Wai et al. [29] achieved the highest DE of pectin extracted from durian rind (65%) by using the following extraction conditions: 80 °C, 1 h, and pH 2.5. The harsh conditions of temperature and pH could increase deesterification of GalA [29]. The L* decreased with the increasing of temperature. This can be explained due to the releasing of phenolic compounds during conventional extraction at a high temperature (90 °C). The increased solvent temperature enhanced the extraction yield and mass transfer rate of different polyphenols (epicatechin, procyanidin, delphinidin, etc.) [43].

### 3.7. FT-IR Analysis

The information on the functional groups presents in the pectin from grape pomace of two varieties Fetească Neagră and Rară Neagră (FN and RN, respectively) extracted in the following conditions: pH 2, particle size of ≥125–<200 µm, 90 °C, and extraction time of 3 h, was evaluated by FT-IR analysis. The attenuated total reflectance FT-IR spectra of grape pomace pectin are illustrated in Figure 3a,b. Irrespective of acid type, pectin samples from FN grape pomace had a similar transmission pattern to those of pectin from RN grape pomace. Grape pomace pectin samples had a characteristic chemical shift at 3310 cm^−1^, which was attributed to intermolecular hydrogen bonding of O(6)H∙∙∙O(3) [55]. Pectin sample extracted with citric acid from FN grape pomace had a peak at 2923 cm^−1^. The presence of this shift was assigned to –CH stretching vibration peak [56]. As well, the pectin sample extracted with sulfuric acid from RN grape pomace had a peak located at around 2360 cm^−1^. This peak was attributed to the C–H bond [57], S–H or C–S [58], which is characteristic of pectin extracted with sulfuric acid [59]. Moreover, this peak may not be regarded as a critical functional group, which did not influence significantly the structure of extracted pectin samples [58].

The FT-IR spectra in the region between 2000 and 1000 cm^−1^ represents the major chemical functional groups in pectin [31,60,61]. The absorption band at around 1717 cm^−1^ was associated with phenolic esters of pectin [61]. The band at around 1560 cm^−1^ corresponded to the protein amide in the pectin molecules [62]. The peak of 1402 cm^−1^ suggested –CH bending of –CH_2_ groups [63]. The peaks between 1320 cm^−1^ and 1000 cm^−1^ showed the presence of alcohols, esters, ethers, and carboxylic acids (C–O stretch) [34]. The band 1304 cm^−1^ was attributed to the asymmetrical stretching of the COO^-^ and C–O group [60]. Skeletal bending modes of H–C–C and C–O–H are predominant in the 1265–1205 cm^−1^. The bands in the region 1200–850 cm^−1^ were produced by stretching vibration of C–O, C–C, and ring structures [64]. Intense peaks in the region 1150–1000 cm^−1^ were as a result of a high amount of homogalacturonan in pectin. A less intense peak at around 1132 cm^−1^ was due to C–O stretching vibrations [64]. The peak located at around 1065 cm^−1^ was assigned to C–O and C–C stretching vibrations [65]. The band positions of 904 cm^−1^ and 842 cm^−1^ corresponded for β-glycosidic linkages and CH_2_ bending, respectively [65]. Peak appearing at around 790 cm^−1^ can be attributed to cyclic –CH bending and to the C-C deformation vibrations of pectin ring [64].

### 3.8. Microstructural Analysis by SEM

Scanning electron microscopy (SEM) is an affective technique in analysis of different materials (organic and inorganic) on a nanometer to micrometer (µm) scale [66]. The morphology of the pectin from grape pomace of two varieties Fetească Neagră and Rară Neagră (FN and RN, respectively) extracted in the following conditions: pH 2, particle size of ≥125–<200 µm, 90 °C, and extraction time of 3 h was investigated by using SEM (Figure 4a,b). As it illustrated, a few changes were established among the samples subjected to the coventional extraction with citric acid (CA), sulfuric acid (SA) and nitric acid (NA) from Fetească Neagră (FN) and Rară Neagră (RN) grape pomace. The pectin extracted with CA from FN had a compact sturcture with long sharped particles (Figure 4ai), while the pectin extracted with CA from RN (Figure 4bi) presented a heterogeneous structure consisted of various rough and irregular surfaces which are compact in shape. This can be explained due to a higher amount of neutral sugar sides in their framework [67]. The pectin samples obtained with SA from both varieties of grape pomace (Figure 4aii,bii) presented a tendency to be curly. In this case, particles showed a discrete elongated structure joined together. The pectin extracted with NA from FN and RN (Figure 4aiii,biii) had an irregular morphology with compact aggregates, i.e., pectin samples presented a nonporous structure. The rough and wrinkled surface of pectin samples can be explained due to high temperature (90 °C) applied in the process of extraction [34,44]. Similar results were obtained by Dalpasquale et al. [68] and Almeida et al. [69]. However, under the traditional extraction, the pectin samples mainly remained as the cluster formation [69].

## 4. Conclusions

The novelty of the present work is that only a few studies are about the pectin extraction from grape pomace. The mechanism of pectin extraction was studied under the following conditions: grape pomace variety, acid type, particle size, extraction temperature, pH, and time. The results showed that extraction conditions have a huge impact on the structure and properties of pectin. The highest yield, GalA, DE, MeO, and EW of pectin from both grape pomace varieties (FN and RN) were obtained for citric acid extraction at pH 2, particle size interval of ≥125–<200 µm, and temperature of 90 °C for 3 h. The obtained results are in agreement with other studies, in which it was established that citric acid at pH 2 is a great alternative to the different mineral acids. Overall, we conclude that grape pomace is not just a waste generated in wine industry, but a by-product with commercially important fibres, such as pectin.

## Figures and Tables

**Figure 1 polymers-14-01378-f001:**
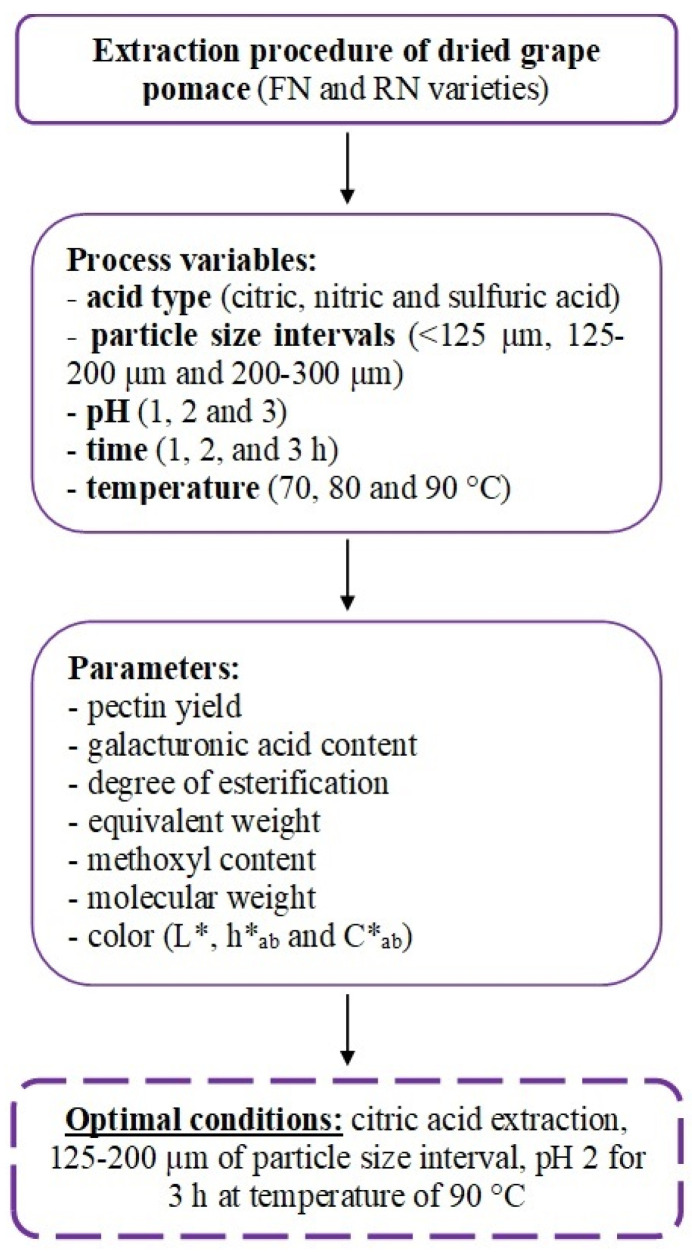
Flow chart representing the extraction procedure in order to establish the optimal conditions.

**Figure 2 polymers-14-01378-f002:**
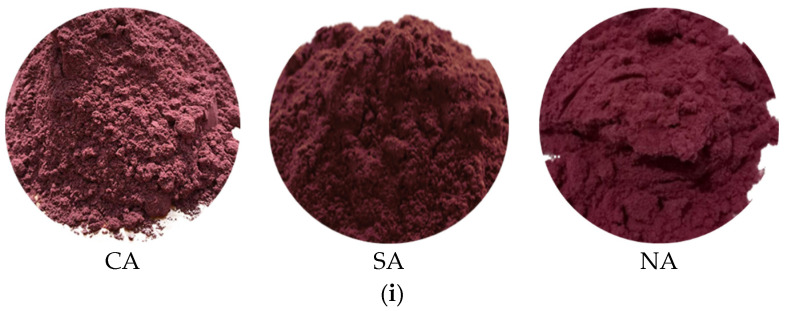
Images of pectin samples extracted from Fetească Neagră and Rară Neagră grape pomace under the influence of acid type: CA—citric acid, SA—sulfuric acid, and NA—nitric acid (fixed extraction conditions: particle size of ≥125–<200 µm, at pH 2, 90 °C for 3 h). (**i**) Pectin samples extracted from Fetească Neagră grape pomace under the influence of acid type. (**ii**) Pectin samples extracted from Rară Neagră grape pomace under the influence of acid type.

**Figure 3 polymers-14-01378-f003:**
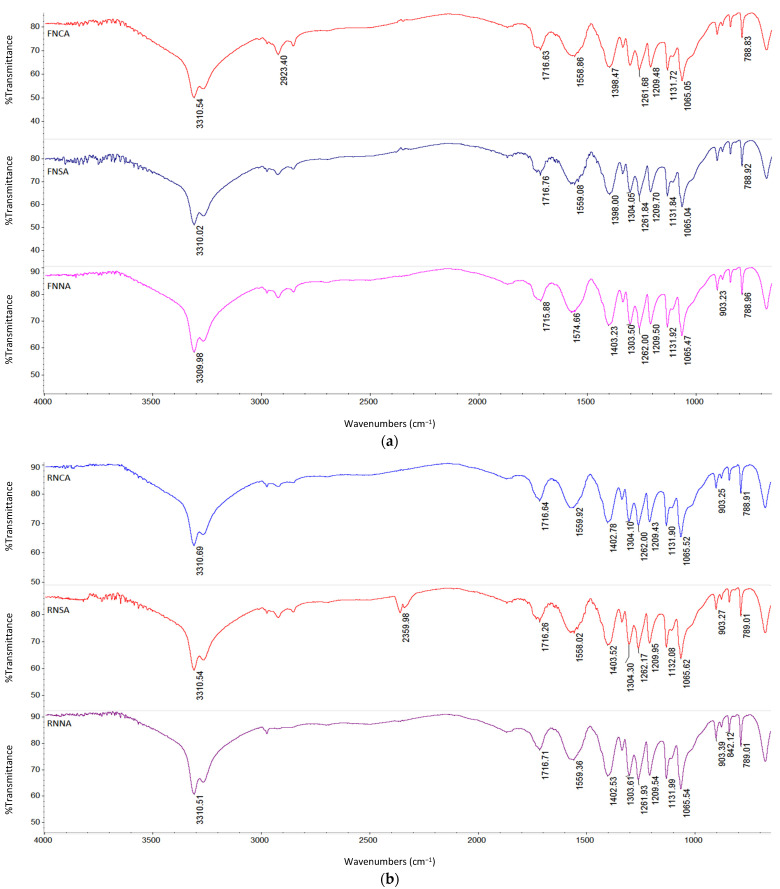
(**a**) FT-IR spectra of pectin samples extracted from Fetească Neagră grape pomace under the influence of acid type: FN—Fetească Neagră, CA—citric acid, SA—sulfuric acid and NA—nitric acid (fixed extraction conditions: particle size of ≥125–<200 µm, at pH 2, 90 °C for 3 h). (**b**) FT-IR spectra of pectin samples extracted from Rară Neagră grape pomace under the influence of acid type: RN—Rară Neagră, CA—citric acid, SA—sulfuric acid and NA—nitric acid (fixed extraction conditions: particle size of ≥125–<200 µm, at pH 2, 90 °C for 3 h).

**Figure 4 polymers-14-01378-f004:**
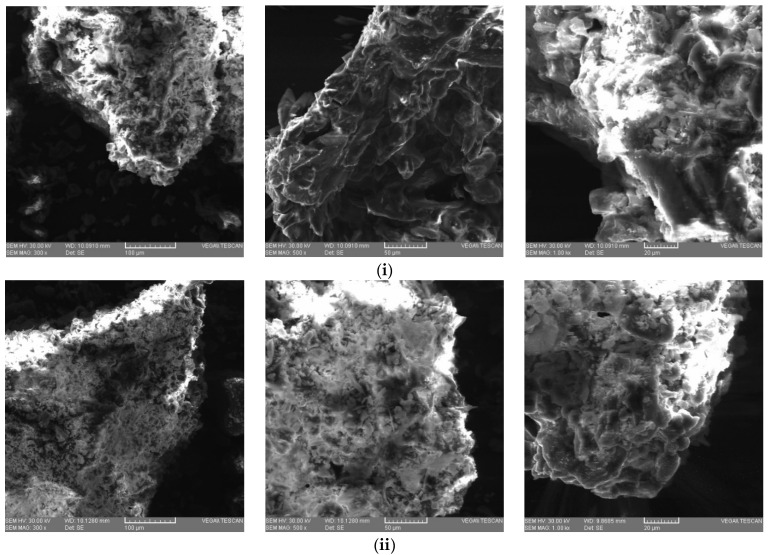
(**a**) SEM images of pectin samples extracted with different acid type from Fetească Neagră grape pomace (fixed extraction conditions: particle size of ≥125–<200 µm, at pH 2, 90 °C for 3 h). (**i**) Pectin samples extracted with citric acid (CA). (**ii**) Pectin samples extracted with sulfuric acid (SA). (**iii**) Pectin samples extracted with nitric acid (NA). (**b**) SEM images of pectin samples extracted with different acid type from Rară Neagră grape pomace (fixed extraction conditions: particle size of ≥125–<200 µm, at pH 2, 90 °C for 3 h). (**i**) Pectin samples extracted with citric acid (CA). (**ii**) Pectin samples extracted with sulfuric acid (SA). (**iii**) Pectin samples extracted with nitric acid (NA).

**Table 1 polymers-14-01378-t001:** Analysis of variance (ANOVA) of the grape pomace variety, acid type, and particle size influence on the yield and physico-chemicals parameters of pectin. Means values and standard deviation are shown in brackets.

Parameter	Grape Pomace Variety		*F*-value	Acid Type			*F*-Value	Particle Size (µm)			*F*-Value
	FN	RN		CA	SA	NA		<125	≥125–<200 µm	≥200–<300	
Yield (%)	6.07 (0.13) ^a^	5.70 (0.21) ^a^	1.23 ^ns^	6.01 (0.18) ^a^	5.80 (0.27) ^a^	5.84 (0.24) ^a^	0.13 ^ns^	4.64 (0.11) ^c^	7.31 (0.10) ^a^	5.71 (0.09) ^b^	147.08 ***
GalA (g/100 g)	33.14 (0.25) ^b^	48.47 (0.27) ^a^	10.67 *	52.01 (0.45) ^a^	31.45 (0.34) ^b^	38.95 (0.38) ^b^	6.76 *	31.55 (0.21) ^b^	56.33 (0.19) ^a^	34.54 (0.18) ^b^	13.96 ***
DE (%)	73.88 (0.20) ^a^	73.15 (0.21) ^a^	0.22 ^ns^	73.21 (0.25) ^a^	74.15 (0.19) ^a^	73.19 (0.27) ^a^	0.16 ^ns^	71.31 (0.42) ^a^	74.31 (0.32) ^a^	74.92 (0.27) ^a^	2.15 ^ns^
EW (g/mol)	555.38 (0.31) ^a^	548.09 (0.21) ^a^	0.38 ^ns^	566.85 (0.09) ^a^	557.32 (0.11) ^ab^	531.04 (0.24) ^b^	3.63 *	549.13 (0.24) ^a^	565.70 (0.15) ^a^	540.38 (0.25) ^a^	1.63 ^ns^
MeO (%)	5.55 (0.31) ^a^	4.30 (0.25) ^b^	12.36 **	5.31 (0.41) ^a^	4.89 (0.27) ^a^	4.56 (0.24) ^a^	1.25 ^ns^	4.60 (0.32) ^b^	6.18 (0.48) ^a^	3.98 (0.25) ^b^	18.57 ***
M_w_ (g/mol)	5.29 × 10^4^ (0.07) ^a^	5.30 × 10^4^ (0.05) ^a^	0.30 ^ns^	5.34 × 10^4^ (0.05) ^a^	5.30 × 10^4^ (0.03) ^b^	5.25 × 10^4^ (0.04) ^c^	10.89 ***	5.29 × 10^4^ (0.04) ^b^	5.35 × 10^4^ (0.08) ^a^	5.24 × 10^4^ (0.03) ^c^	17.09 ***
L*	41.62 (0.24) ^b^	49.14 (0.33) ^a^	21.18 ***	48.33 (0.12) ^a^	47.34 (0.15) ^a^	40.44 (0.17) ^b^	8.68 **	47.78 (0.21) ^a^	45.98 (0.28) ^ab^	42.35 (0.17) ^b^	2.99 ^ns^
h*_ab_	31.06 (0.25) ^b^	39.27 (0.29) ^a^	32.33 ***	32.90 (0.14) ^b^	34.77 (0.09) ^ab^	37.82 (0.10) ^a^	2.63 ^ns^	36.80 (0.17) ^a^	34.60 (0.24) ^a^	34.08 (0.27) ^a^	0.84 ^ns^
C*_ab_	10.89 (0.14) ^a^	11.26 (0.12) ^a^	0.30 ^ns^	11.23 (0.17) ^ab^	10.16 (0.23) ^b^	11.84 (0.18) ^a^	2.37 ^ns^	10.48 (0.03) ^a^	10.93 (0.05) ^a^	11.81 (0.02) ^a^	1.45 ^ns^

ns—*p* > 0.05, *—*p* < 0.01, **—*p* < 0.001, ***—*p* < 0.0001, ^a–c^—different letters in the same row indicate significant differences between samples (*p* < 0.0001) according to the LSD test with α = 0.05. FN—Fetească Neagră, RN—Rară Neagră, GalA—galacturonic acid content, DE—degree of esterification, EW—equivalent weight, MeO—methoxyl content, M_w_—molecular weight. Extraction conditions—pH 2, temperature of 90 °C for 3 h.

**Table 2 polymers-14-01378-t002:** Analysis of variance (ANOVA) of the pH influence on the yield and physico-chemical parameters of pectin. Means values and standard deviation are shown in brackets.

Parameter	Grape Pomace Variety		*F*-Value	pH			*F*-Value
	FN	RN		1	2	3	
Yield (%)	9.56 (0.12) ^a^	7.87 (0.18) ^a^	1.29 ^ns^	12.43 (0.32) ^a^	7.56 (0.25) ^b^	6.14 (0.27) ^b^	23.89 ***
GalA (g/100 g)	42.86 (0.56) ^a^	54.96 (0.47) ^a^	4.23 ^ns^	39.40 (0.17) ^b^	61.76 (0.21) ^a^	45.57 (0.19) ^b^	7.73 *
DE (%)	71.67 (0.33) ^a^	75.39 (0.37) ^a^	3.46 ^ns^	70.39 (0.18) ^b^	78.92 (0.13) ^a^	71.27 (0.16) ^b^	23.30 ***
EW (g/mol)	563.76 (0.11) ^a^	549.41 (0.13) ^a^	1.43 ^ns^	559.26 (0.34) ^b^	582.97 (0.38) ^a^	527.51 (0.41) ^c^	34.48 ***
MeO (%)	6.77 (0.32) ^a^	6.72 (0.25) ^a^	0.19 ^ns^	6.71 (0.07) ^b^	6.99 (0.08) ^a^	6.53 (0.06) ^c^	33.53 ***
M_w_ (g/mol)	5.35 × 10^4^ (0.20) ^a^	5.34 × 10^4^ (0.22) ^a^	0.06 ^ns^	5.39 × 10^4^ (0.19) ^a^	5.37 × 10^4^ (0.16) ^b^	5.28 × 10^4^ (0.21) ^c^	1116.95 ***
L*	39.20 (0.11) ^a^	46.80 (0.05) ^a^	1.02 ^ns^	23.89 (0.14) ^c^	45.20 (0.12) ^b^	59.92 (0.04) ^a^	76.12 ***
h*_ab_	32.03 (0.24) ^a^	35.86 (0.21) ^a^	0.86 ^ns^	25.06 (0.35) ^c^	34.28 (0.38) ^b^	42.48 (0.41) ^a^	17.86 **
C*_ab_	10.50 (0.02) ^a^	9.39 (0.04) ^a^	0.30 ^ns^	4.41 (0.21) ^b^	13.06 (0.09) ^a^	12.37 (0.17) ^a^	114.04 ***

ns—*p* > 0.05, *—*p* < 0.01, **—*p* < 0.001, ***—*p* < 0.0001, ^a–c^—different letters in the same row indicate significant differences between samples (*p* < 0.0001) according to the LSD test with α = 0.05. FN—Fetească Neagră, RN—Rară Neagră, GalA—galacturonic acid content, DE—degree of esterification, EW—equivalent weight, MeO—methoxyl content, M_w_—molecular weight. Extraction conditions—citric acid, ≥125–<200 µm of particle size, temperature of 90 °C for 3 h.

**Table 3 polymers-14-01378-t003:** Analysis of variance (ANOVA) of the time influence on the yield and physico-chemicals parameters of pectin. Means values and standard deviation are shown in brackets.

Parameter	Grape Pomace Variety		*F*-Value	Time (h)			*F*-Value
	FN	RN		1	2	3	
Yield (%)	6.05 (0.06) ^a^	6.37 (0.12) ^a^	0.46 ^ns^	5.38 (0.21) ^c^	5.77 (0.18) ^b^	7.50 (0.17) ^a^	187.94 ***
GalA (g/100 g)	38.39 (0.05) ^a^	41.41 (0.11) ^a^	0.84 ^ns^	33.69 (0.37) ^b^	39.04 (0.28) ^b^	46.98 (0.31) ^a^	14.01 **
DE (%)	74.79 (0.11) ^a^	72.91 (0.17) ^a^	0.61 ^ns^	67.17 (0.13) ^b^	76.61 (0.11) ^a^	77.77 (0.12) ^a^	140.90 ***
EW (g/mol)	613.41 (0.14) ^a^	586.95 (0.17) ^b^	8.66 *	586.20 (0.15) ^b^	595.92 (0.14) ^ab^	618.42 (0.13) ^a^	4.31 *
MeO (%)	5.70 (0.11) ^b^	6.86 (0.14) ^a^	75.02 ***	6.38 (0.07) ^a^	5.93 (0.11) ^a^	6.54 (0.09) ^a^	1.50 ^ns^
M_w_ (g/mol)	5.30 × 10^4^ (0.20) ^a^	5.31 × 10^4^ (0.24) ^a^	0.20 ^ns^	5.25 × 10^4^ (0.19) ^c^	5.38 × 10^4^ (0.17) ^a^	5.29 × 10^4^ (0.23) ^b^	407.77 ***
L*	67.55 (0.13) ^a^	63.96 (0.17) ^a^	2.24 ^ns^	59.98 (0.34) ^c^	71.52 (0.27) ^a^	65.76 (0.24) ^b^	41.71 ***
h*_ab_	42.44 (0.28) ^a^	41.14 (0.33) ^a^	0.19 ^ns^	33.62 (0.15) ^b^	46.71 (0.14) ^a^	45.06 (0.17) ^a^	110.21 ***
C*_ab_	14.04 (0.16) ^a^	13.29 (0.24) ^a^	1.47 ^ns^	12.82 (0.07) ^b^	14.62 (0.09) ^a^	13.57 (0.11) ^ab^	3.58 ^ns^

ns—*p* > 0.05, *—*p* < 0.01, **—*p* < 0.001, ***—*p* < 0.0001, ^a–c^—different letters in the same row indicate significant differences between samples (*p* < 0.0001) according to the LSD test with α = 0.05. FN—Fetească Neagră, RN—Rară Neagră, GalA—galacturonic acid content, DE—degree of esterification, EW—equivalent weight, MeO—methoxyl content, M_w_—molecular weight. Extraction conditions—citric acid at pH 2, ≥125–<200 µm of particle size, temperature of 90 °C.

**Table 4 polymers-14-01378-t004:** Analysis of variance (ANOVA) of the temperature influence on the yield and physico-chemical parameters of pectin. Means values and standard deviation are shown in brackets.

Parameter	Grape Pomace Variety		*F*-Value	Temperature (°C)			*F*-Value
	FN	RN		70	80	90	
Yield (%)	5.74 (0.24) ^a^	5.96 (0.32) ^a^	0.12 ^ns^	4.23 (0.34) ^c^	6.02 (0.27) ^b^	7.30 (0.17) ^a^	494.78 ***
GalA (g/100 g)	53.27 (0.18) ^a^	48.40 (0.21) ^a^	1.01 ^ns^	39.16 (0.27) ^c^	51.14 (0.32) ^b^	62.21 (0.36) ^a^	56.00 ***
DE (%)	84.98 (0.14) ^a^	70.90 (0.17) ^b^	92.08 ***	74.65 (0.05) ^a^	77.77 (0.08) ^a^	81.40 (0.11) ^a^	1.13 ^ns^
EW (g/mol)	535.38 (0.05) ^a^	569.06 (0.07) ^a^	2.46 ^ns^	511.44 (0.21) ^b^	536.96 (0.16) ^b^	608.30 (0.14) ^a^	28.33 ***
MeO (%)	6.34 (0.02) ^a^	6.38 (0.06) ^a^	0.02 ^ns^	5.60 (0.12) ^b^	6.74 (0.14) ^a^	6.75 (0.16) ^a^	578.00 ***
M_w_ (g/mol)	5.28 × 10^4^ (0.17) ^a^	5.29 × 10^4^ (0.21) ^a^	0.04 ^ns^	5.23 × 10^4^ (0.15) ^b^	5.31 × 10^4^ (0.20) ^a^	5.32 × 10^4^ (0.23) ^a^	14.60 **
L*	52.40 (0.07) ^a^	53.67 (0.12) ^a^	0.20 ^ns^	59.72 (0.24) ^a^	50.74 (0.21) ^b^	48.64 (0.19) ^b^	17.15 **
h*_ab_	24.93 (0.03) ^b^	37.90 (0.05) ^a^	37.90 ***	26.83 (0.06) ^a^	34.74 (0.11) ^a^	32.67 (0.12) ^a^	1.74 ^ns^
C*_ab_	12.14 (0.09) ^a^	14.48 (0.11) ^a^	1.57 ^ns^	8.37 (0.19) ^c^	14.45 (0.18) ^b^	17.13 (0.21) ^a^	50.95 ***

ns—*p* > 0.05, *—*p* < 0.01, **—*p* < 0.001, ***—*p* < 0.0001, ^a–c^—different letters in the same row indicate significant differences between samples (*p* < 0.0001) according to the LSD test with α = 0.05. FN—Fetească Neagră, RN—Rară Neagră, GalA—galacturonic acid content, DE—degree of esterification, EW—equivalent weight, MeO—methoxyl content, M_w_—molecular weight. Extraction conditions—citric acid at pH 2, ≥125–<200 µm of particle size, for 3 h.

## Data Availability

Not applicable.

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
