# Peer review of "The Influence of Extraction Conditions on the Yield and Physico-Chemical Parameters of Pectin from Grape Pomace"

_polymers, 2022, doi:10.3390/polym14071378_

Round 1

Reviewer 1 Report

In this paper, the authors described and discussed the influence of the acid type (organic vs. mineral), temperature, pH, particle size and time on the yield and physico-chemical parameters (galacturonic acid content, degree of esterification, methoxyl content, molecular and equivalent weight) of pectin from grape pomace of two different Vitis vinifera varieties (Fetească Neagră and Rară Neagră) using conventional method of extraction and establish the optimal conditions of grape pomace pectin extraction.. The paper fit the aims and scope of Polymers. I would recommend accepting the paper after modifications. I have some comments to the authors.

  1. The novel utilization of grape pomace could realize the high-value utilization of wastes or by-products.Some related studies on the extraction of high value products from waste or by-products should be mentioned. doi: 10.1016/j.lwt.2021.111617, doi:10.3390/foods9040449, doi: 10.3390/polym13152578, doi:1002/jsfa.11055, doi: 10.3390/pr7120875
  2. Line 99: centrifugation at 2,320 g, why at 2,320 g?How was such a precise parameter decided?
  3. Section 2.2.7 should be described with more detail.
  4. The author useda lot of examples in the Section Results and Discussion, but did not seem to have a good description of what to discuss through these examples.

Reviewer 2 Report

Overall, this manuscript presents simple information of pectin extraction for grape pomace. Pectin extraction has been widely investigated in several raw materials. As mentioned by Authors, pectin extracted from grape pomace have gone so far with other advanced techniques, while this manuscript presents conventional extraction.Thus, the novelty is low.

Keywords, influence must be deleted. It is too simple.

Too many references were cited. Again, several references were too old.

Therefore, I recommend Reject.

Reviewer 3 Report

The paper is very interesting, well written and well organized. In addition, there is only a few studies about the pectin extraction from grape pomace. The ways and means are well described as well as the obtained results which are statistically analyzed. However, in order to make it a stronger paper, I suggest some points before it be published:

  1. What is the theoretical pectin content on grape pomace? Pectin yields could be better if the real grape pomace content of pectin was considered in Equation 1 to calculate extraction yield.
  2. Line 126: 0.1 N is not very usual anymore, please replace by the mol/L content.
  3. As authors explored the pectin color, it would be interesting to add a picture of the produced pectin.
  4. Authors says that “The grape pomace pectin yield presented the following values, 4.64%, 5.71% and 7.31% for particle size intervals of <125 μm, ≥200 - <300 μm and ≥125 - <200 μm, respectively”, however all cited references had the pectin yield improved with smaller particles (with makes sense according to surface area contact and mass transfer limitation). How authors explain the opposite behavior? In this work the higher yield was achieved with the middle size particles.
  5. The manuscript has more than 90 references, occupying a lot of pages. This is generally the number of references in review papers. Does authors consider all of those references really relevant? I believe some of then could be removed.

Round 2

Reviewer 2 Report

The revised manuscript has been improved and some evidence proposed by Authors can be accepted. Thus, I recommend Accept.

This manuscript is a resubmission of an earlier submission. The following is a list of the peer review reports and author responses from that submission.

Round 1

Reviewer 1 Report

Comments to the authors Manuscript ID: polymers-1397147 Title: The influence of extraction conditions on the yield and physico-chemical parameters of pectin from grape pomace. The manuscript presents the influence of extraction conditions on the yield and physico-chemical parameters of grape pomace pectin. However, I consider that the authors should reconsider the novelty and contribution of this work to the knowledge because there is a lot of information into the extraction conditions on the pectin structure and properties. Also, the authors omitted some important measurements on the pectin characterization (neutral monosaccharide composition by GLC, uronic acid content by HPAEC-PAD, molar mass distribution by HPSEC or HPGPC, structural properties by HSQC-NMR, protein content, ash content, DSC analysis, antioxidant activity, and rheology parameters).

Reviewer 2 Report

Review on manuscript ID: polymers-1397147 “The influence of extraction conditions on the yield and  physico-chemical parameters of pectin from grape pomace” by Mariana Spinei  and Mircea Oroian submitted to Polymers

The article presented is well structured.The aim of studies is to clearly formulate and correctly select the analytical methods necessary for its implementation.

In my opinion the topic taken by Authors is interesting. Generally manuscript is readable , prepared correctly and could be publish in Polymers after minor corrections.

Detailed recommendations:

Page 2, line 65

- The authors report that they used grape pomace with three particle size ranges, i. e. tj.: < 125 µm, 125 – 200 µm i 200 – 300 µm for the preparation of pectins.

 In my opinion, the particle size ranges of 125-200 µm and 200-300 µm are not entirely correct. It would be correct if they wrote: ≥125 - < 200 µm i ≥200 - < 300 µm.

Page 2, line 80 and 85

- In my opinion, the authors should make corrections in the descriptions of the centrifugation conditions, because now, instead of rpm, the value of the centrifugal force acting on the sample is given, which depends on the rotor geometry and the acceleration due to gravity (g) and the time necessary for the centrifugation process.

- In my opinion, the authors should indicate the temperature at which the samples were centrifuged.

Page 3, lines 93 – 96

Please detail the methodology for the determination of galacturonic acid in pectins.

Page 3, line 124

The authors write that „The mixed solution was stirred thoroughly and kept for 30 min at room temperature.” Please specify exactly at what temperature this solution was kept?

Page 9

In table 3, in the header the letter h should be written in brackets.

Page 17, line 690

There are no page numbers in the listed references.

Page 18, line 700

There are no page numbers in the listed references.

Page 18, line 709

Remove the dashes between letters.

Reviewer 3 Report

In the manuscript entitled “The influence of extraction conditions on the yield and 2 physico-chemical parameters of pectin from grape pomace”, the authors studied the influence of extraction conditions on the yield and physico-chemical properties of pectin from grape pomace, a by-product of the wine industry. The paper fit the aims and scope of Polymers. I would recommend accepting the paper after addressing several issues below.

  1. Abstract should be revised to clarify that specific physical and chemical properties were investigated
  2. The author should explain why it is interesting to do the experiments they describe and especially what is new compared to these published papers. In my opinion, source of raw materials which can be obtained from agricultural and industrial processing by-products and wastes, is worth to be emphasized. Extracting pectin from waste and reusing it should be a highlight of this article, which was contributed to establish a circular economy to minimize resource use and waste. In order to emphasize this point, the author should further introduce the situation in reuse of waste. The research status of the reuse of wastes, not limited to solid wastes, should be introduced. Therefore, the incipit has to be supported with proper suitable literature references, in order to evidence the need to reuse agricultural and industrial waste and by-products, especially the separation of bioactive substances from agricultural and industrial waste and by-products.

doi: 10.1016/j.lwt.2021.111617, doi: 10.1002/jsfa.11055, doi: 10.1016/j.ijbiomac.2018.02.018, doi: 10.3390/polym13081280.

  1. It was strongly suggested to indicate at the end of the Introduction section the main employed characterisation techniques in order to achieve their purpose.
  2. Line63-66: Was one sample divided into three parts according to the degree of grinding, or three identical samples were grinded to different particle sizes respectively?
  3. Line 85: What did “this” mean?
  4. Section 2.2.3 should be revised with more details.
  5. Line 101: 0.1 N should be changed to SI units.
  6. The abbreviation of “equivalent weight” was recommended to be changed to Weq or EW to avoid confusion with the abbreviation of “equation”.
  7. Figure 1 was suggested to be moved to Section 2.
  8. The difference at about 2360 cm-1 in Figure 2.b should be explained.

Reviewer 4 Report

In this paper, the author investigated the influence of extraction conditions on the fields and physicochemical parameters of pectin from grape pomace, which is the by-product generated from wine industry and could be made a better use of. 

The folowing conditions, such as grape pomace variety (Fetească Neagră and Rară Neagră), acid type (citric, sulfuric and nitric acid), particle size intervals (<125 μm, 125-200 μm and 200-300 μm), temperature (70, 80 and 90°C), pH (1, 2 and 3) and extraction time (1, 2 and 3 h) were carefully investigated so as to optimize the extraction conditions of pectin.

Results showed that acid type, particle size intervals, temperature, time and pH had significant influence on the yield and physicochemical parameters of pectin extracted from grape pomace.

Extraction with citric acid at pH 2, particle size interval of 125-200 μm, temperature of 90 °C for 3 h turned out to be the optimal extraction conditions to obtain the highest yield, galacturonic acid content, degree of esterification, methoxyl content and equivalent weight of pectin. The utilization of organic acid (i.e., citric acid) also meets the developing concept of green chemistry and technology.

Questions:

  • Page 3, line 102, typo continuos --> continuous
  • In the section 3.2 (page 6), the author discussed and compared the yields and physicochemical parameters of pectin extracted by different acids in this work and other cited works. These data and numbers are very helpful and can give a clear comparison to readers. It would be even more helpful if some more discussion and explanation could be added to demonstrate the possible/potential reason why organic acid is better than mineral acids? Is there any fundamental chemistry or other science behind it could explain it rather than simply list all the numbers.    
  • In the section 3.3 (page 7), the author talked about the influence of particle size on the yield and physicochemical parameters of grape pomace. The yield, GalA and MeO turned to be significantly influenced by the particle size of grape pomace. The author also mentioned pectin from sugar beet pulp (ref 52 and 54) and pectin from melon peel (ref 43) indicated that higher yields received for smaller particle sizes. However, the results in this work seemed not follow the same trend. It would be helpful if the author could discuss a little bit more about the potential reason or hypothesis why grape pomace behaved differently rather than only showed the values to readers.
  • In the section 3.7 (page 11), the author mentioned that two unique FT-IR peaks (2923 cm-1 from CA extraction of FN and 2360 cm-1 from SA extraction of RN) were observed in the FT-IR spectra. I'm wondering whether the author could comment about this. Does that mean that the some characteristic pectin from different variety of grape pomace need be extracted out by using specific acid?